# Developments in Rabies Vaccines: The Path Traversed from Pasteur to the Modern Era of Immunization

**DOI:** 10.3390/vaccines11040756

**Published:** 2023-03-29

**Authors:** Krithiga Natesan, Shrikrishna Isloor, Balamurugan Vinayagamurthy, Sharada Ramakrishnaiah, Rathnamma Doddamane, Anthony R. Fooks

**Affiliations:** 1KVAFSU-CVA Rabies Diagnostic Laboratory, WOAH Reference Laboratory for Rabies, Department of Veterinary Microbiology, Veterinary College, KVAFSU, Hebbal, Bengaluru 560024, Karnataka, India; 2ICAR-NIVEDI, Yelahanka, Bengaluru 560064, Karnataka, India; 3APHA Weybridge, Woodham Lane, New Haw, Addlestone, Surrey KT15 3NB, UK

**Keywords:** rabies, history, vaccines, reverse genetics

## Abstract

Rabies is a disease of antiquity and has a history spanning millennia ever since the first interactions between humans and dogs. The alarming fatalities caused by this disease have triggered rabies prevention strategies since the first century BC. There have been numerous attempts over the past 100 years to develop rabies vaccineswith the goal of preventing rabies in both humans and animals. Thepre-Pasteurian vaccinologists, paved the way for the actual history of rabies vaccines with the development of first generation vaccines. Further improvements for less reactive and more immunogenic vaccines have led to the expansion of embryo vaccines, tissue culture vaccines, cell culture vaccines, modified live vaccines, inactivated vaccines, and adjuvanted vaccines. The adventof recombinant technology and reverse genetics have given insight into the rabies viral genome and facilitated genome manipulations, which in turn led to the emergence of next-generation rabies vaccines, such as recombinant vaccines, viral vector vaccines, genetically modified vaccines, and nucleic acid vaccines. These vaccines were very helpful in overcoming the drawbacks of conventional rabies vaccines with increased immunogenicity and clinical efficacies. The path traversed in the development of rabies vaccines from Pasteur to the modern era vaccines, though, faced numerous challenges;these pioneering works have formed the cornerstone for the generation of thecurrent successful vaccines to prevent rabies. In the future, advancements in the scientific technologies and research focus will definitely lay the path for much more sophisticated vaccine candidates for rabies elimination.

## 1. Introduction

Rabies has a historical importance ever since the beginning of human and dog relationships nearly 40,000 years ago. The Mesopotamian records reveal the existence of a very hazardous ‘mad dog’ disease, which reveals the interaction of dogs with a most deadly rabies virus [1]. Rabies, a fatal infectious, zoonotic disease, is caused by the rabies virus (RABV) [2]. The disease continues to pose a serious threat to global public health, particularly in developing countries. This acute progressive encephalitis claims approximately 60,000 human fatalities annually, with its major toll in Africa (36.4%) and Asia (59.6%) [3]. Among which, South Asia contributes to about 40% of the total human rabies mortalities in the world. Estimates place the cost of rabies at US$583.5 million annually, with livestock losses in Asia and Africa costing about US$12.3 million. Canine rabies is present in 87 different nations and is the main factor in cases of human rabies. However, several nations, including Japan, the United Kingdom, Denmark, Sweden, Greece, Ireland, Iceland, Portugal, New Zealand, Australia, Switzerland, Finland, Norway, France, and Belgium, among others, have eradicated rabies [4].

The main perpetuator of the disease is the rabies virus (RABV), a type species of the genus Lyssavirus in the family *Rhabdoviridae.* It is a bullet shaped virus, holding a single-stranded, negative-sense RNA genome of about 12 kb, which encodes five major structural proteins from 3′ to 5′ viz., nucleoprotein (N), phosphoprotein (P), matrix protein (M), glycoprotein (G), and RNA-dependent RNA polymerase (L) [5]. The N, P, and L proteins create a ribonucleoprotein complex that tightly encapsidates the negative-sense RNA genome and is in charge of directing viral replication in the cytoplasm of infected cells. The RABV G protein is the sole viral protein that is exposed on the surface of the virus and serves as a major factor contributing to viral pathogenicity and acts as a primary protective antigen resulting in protective immunity against rabies [6].

The disease affects all warm-blooded animals, including humans, and rabies virus has extended its host range within the mammalian orders *Carnivora* and *Chiroptera* [7]. However, among them, dogs are the most important domestic reservoir hosts for human infectionsin developing countries, whereas wildlife animals serve as hosts in developed countries [8]. Apart from dogs, several species of bats, especially vampire bats, also play a crucial role in the transmission of rabies virus in humans in the American continent [9]. In contrast, lyssavirus species are transmitted by bats in the Old World countries in Africa, Asia, and Europe [10]. Other domestic animals, including cats, cattle, horses, sheep, and goats, could contract rabies and spread it across to humans [11]. Infected dog bites account for 97% of human rabies cases, followed by cat bites (2%), and other animal bites (1%), including those from mongoose, fox, wolf, jackal, and other wild animals [12].

Fortunately, rabies vaccines have emerged as the most effective tool to prevent infection by this fatal viral zoonosis. Rabies vaccines can be administered both prophylactically and therapeutically [10], and current vaccines are more efficacious if they are administered in a timely fashion after exposure to rabies. Post-exposure prophylaxis (PEP), involving cleaning the wound at the RABV exposure site, administering rabies immunoglobulins (RIG) if necessary, and administering multiple doses of the rabies vaccine, or the Pre-exposure prophylaxis (PrEP), administering numerous doses of the rabies vaccination prior to exposure to (RABV), were the two major immunization regimens suggested by the World Health Organization (WHO) for the prevention of human rabies [13]. The Global Strategic Plan for the eradication of human rabies deaths caused by dogs worldwide by 2030 was introduced in 2018 by the World Health Organization (WHO), the World Organization for Animal Health (WOAH), the Food and Agriculture Organization of the United Nations (FAO), and the Global Alliance for Rabies Control (GARC). It places a strong emphasis on the prevention of canine rabies through yearly mass vaccination that reaches at least 70% of dog populations [13].

More than a century has passed since Louis Pasteur developed the first vaccine on 6 June 1885, for pre-exposure immunization and post-exposure prophylaxis. Further, several rabies vaccinations have been developed over years and are now being used to prevent or control rabies in humans and animals. Many countries across the world utilize inactivated rabies vaccines based on cell culture; however, these vaccines need multiple inoculations to produce a strong humoral immune response and are more expensive for use to immunize people and animals in developing nations [14]. Although inactivated nerve tissue vaccinations are comparably less expensive, they have been phased out in a number of countries due to their negative side effects, such as neuro-paralytic complications in certain individuals [15]. The attenuated live vaccines using SAD-Bern, Evelyn Rokitnicki Abelseth (ERA), and SAD-B19 strains can effectively elicit a protective immune response with a smaller amount of virus, but still occasionally have the potential to cause rabies in animals due to the virus’s residual virulence or pathogenic mutations during viral propagation in the host [16].

The further need for safe, less reactive, and more immunogenic vaccines resulted in the development of next-generation vaccines. The present understanding of the biology of RABV has been dramatically improved through the introduction of reverse genetics technology and genetic manipulations in terms of recombinant DNA technology [17]. This has also significantly sped up the creation of innovative vaccinations, which has created a platform for next-generation vaccines. The initial genetically modified vaccines altered the rabies viral genome by deleting genes encoding the phosphoprotein or the matrix protein that rendered an apathogenic vaccine virus devoid of neurovirulence even in immunocompromised mice. Genetically modified vaccines, such as rERAG333E, ERAG3G, and SPBN GAS GAS strains, hold a special place in eliciting a remarkable immune response [18,19]. Due to safety concerns, the further variation of recombinant vaccines was performed by manipulating the rabies virus genome to encode two copies of the glycoprotein [20] or expressing only the rabies virus glycoprotein (RAVG) in terms of nucleic acid vaccines, such as the pCIneo plasmid encoding the RABV G protein or the mRNA-based rabies vaccine SFV-RVGP [21,22]. However, these vaccine could not hold an appreciable immune response. Further exploitation of less pathogenic viruses as carrier molecules for RAVG has resulted in viral vector vaccines, which has provided a promising platform for oral vaccinations in wild animals [23]. Currently, the viral vector-based vaccines RABORAL V-RG and ONRAB are much appreciated in controlling wildlife rabies [24,25]. The current research on the introduction ofintradermal rabies vaccination also represents a paradigm shift for post-exposure prophylaxis in Asia [26].

This comprehensive reviewgives an overview of the path traversed in the genesis of rabies vaccines from the paradigm of myths and how the path has moved ahead during the actual history of rabies vaccines with the development of different types of conventional vaccines. In addition, the review emphasizes the emergence of the current vaccines with advanced genetic techniques and also highlights the future focus of rabies vaccine research in the elimination of rabies. Furthermore, this review provides comprehensive information and highlights various types of anti-rabies vaccines used in different countries, their advantages, limitations, and success stories, as well as the failures, which will enable us to design control strategies for the “Control of dog mediated human rabies by 2030”.

## 2. History of Rabies Vaccines

The history of rabies prevention began in the first century BC with many myths and dogma regarding rabies and its treatment. There was no consistent diagnosis or treatment for rabies in humans or animals until the nineteenth century. Techniques, including cauterization, were suggested for the treatment of rabies wounds, and in some cases, they followed even excision or amputation. However, all of these convictions have never been a solution for the alarming fatalities among humans and animals. During 25 AD, people started viewing rabies from a scientific perspective. Aulus Cornelius Celsus in 25 AD promoted the early treatment of wounds after a bite. In 1198, Moses Maimonides portrayed long incubation periods in bitten individuals. Later Giovanni Battista Morgani identified the rabies virus predilection in nerve tissues in 1769. In 1804, Georg Gottfried Zink demonstrated that the infectious saliva from a rabid animal could be the source of infection. In 1852, a French pharmacist, Apollinaire Bouchardat was the first scientist to contemplate the probability of inoculations against rabies infections. In 1881, the first experimental immunization against rabies was achieved in sheep through the intravenous inoculation of rabies virus by a French veterinarian Pierre-Victor Galtier [27].

Although the history of vaccine development was initiated by thepre-Pasteurian vaccinologists, the actual history of rabies vaccine development started in 1885 by Louis Pasteur as an emergency management, even before the causative agent of the disease was identified [28]. Initially, the etiology of rabies did not fit Koch’s Germ Theory as they could not cultivate any infectious agents related to the disease. Even in the late 1800s, rabies was believed to be caused by a parasite, comparable to the Sporozoa [29]. They could not prove any “filterable agent” that causes the disease until 1903. The size of the rabies virion was not resolved until 1936, and an electron microscopic exploration of the causal agent did not occur until 1962. Despite these limitations, in 1881, Pasteur and his team with Chamberland, Roux, and Thuillier could track the presence of the rabies virus in the central nervous system of rabid animals [30]. 

## 3. First Generation Vaccines: Pasteur Vaccine (Nerve Tissue Vaccine)

The era of first-generation vaccines started with Louis Pasteur who developed the first rabies vaccine, from an infected rabbit spinal cord through physical inactivation of the rabies virus via sun drying. Through several passages and the adaptation of street (wild-type) rabies virus to laboratory animals, Pasteur was able to change the virus properties in terms of virulence and the incubation period. Through repeated passaging more than 50 times by inoculating steady amounts of a street virus preparation onto rabbits dura mater membrane, Pasteur observed that the consistency of the incubation period from inoculation to the expansion of rabies was fixed by 7 days. Thus, he termed the virus as a “Fixed” virus. After several experimentations on dogs as the natural hosts, on 6 July 1885, Pasteur first administered his experimental rabies vaccine to a 9-year-old boy, Joseph Meister, who had multiple severe rabid dog bites. Around 2 days post-bite, the little boy received 13 injections of air dried rabies virus-infected, rabbit spinal cord suspensions of progressively increasing virulence for 11 days. This strategic vaccination by Pasteur saved Meister from a rabid death bed [31]. However, the major drawback with the Pasteur vaccine was it contained increasingly virulent rabies virus. Moreover, there was concern with the consistency of inactivation, as few cases were reported with individuals developing rabies post-vaccination. Moreover, the inability to achieve sufficient vaccine production to meet the demand was the main challenge in providing large-scale vaccine production. However, Pasteur’s method was subsequently used for more than a period of 50 years, before significant modifications were introduced in the rabies vaccine preparation. 

## 4. Chemically Modified Fermi & Semple Vaccines 

Pasteur’s vaccine was further improved through a simple chemical modification made by Fermi in 1908 [32] and Semple in 1911 [33]. Newer nerve tissue vaccines (NTVs) were developed by Sir David Semple at the Central Research Institute (CRI), Kasauli, India, from adult sheep (Semple vaccine). They inactivated infected sheep or goat brain with chemical agents, such as phenol [33]. The addition of phenol, although inactivated Pasteur vaccine, still distorted the protein structure and disrupted the rabies virus antigenicity. Moreover, severe side effects, such Guillain-Barre Syndrome (GBS) and the danger of transmitting Transmissible Spongiform Encephalopathies (TSE), were reported. Although this vaccine was widely used in many parts of the world, the WHO eventually suspended its use in almost all countries.

## 5. Myelin-Free Tissue Vaccines 

Though Fermi and Semple vaccines were successful, sensitization in some vaccinated individuals and also the few cases intensified by fatal encephalitis due to the high levels of myelin necessitated an alternative, less reactogenic vaccine. In the 1940s, clinical research on vaccination-related allergic encephalomyelitis and demyelinating diseases in the CNS received significant attention. Later, the advent of embryonated eggs, such as chick or duck embryos and neonatal rodent brains, as the media to produce rabies vaccines has made the journey of vaccine development much safer. Clinical evidence revealed the absence of substances accountable for the vaccine side effects in embryonic and newborn animal nerve tissues. Researchers from the former Soviet Union developed a neonatal rodent brain vaccine using rats [34]. In 1964, Fuenzalida and his team developed a myelin-free inactivated rabies vaccine from the brain of a suckling mouse (Suckling mouse brain-SMB) via phenolic inactivation followed by its partial purification [35].

The lack of myelin in tissues derived from newborn animals made the SMB vaccine less reactogenic compared to the Semple vaccine. However, research revealed that the vaccine was not free from myelin completely, and the presence of other undesirable components resulted in severe adverse reactions [36]. Hence, in unison with WHO recommendations, the national regulatory authorities across the globe decided to discontinue this vaccine after its decade’s long use [37].

## 6. Embryo Vaccines 

In 1931, Ernest W. Goodpasture’s adaptation of various human viruses using embryonated eggs provided a novel platform for the further improvisation of rabies vaccines. Following the use of live animals, embryonated eggs were used in developing rabies vaccines. In 1940, the rabies virus Flury strain was applied to 1-day-old chicks [38]. The strain was subsequently established in chick embryos [39]. The Flury low egg passage (LEP) vaccine was made up of live attenuated virus that had undergone 40–50 egg passages and further lyophilized from a 33% whole-embryo suspension. The LEP vaccine was used in mass dog vaccinations but still had some residual virulence, particularly in kittens, cats, and cattle. Following this, the Flury high egg passage (HEP) vaccine was produced through series of nearly 180 egg passagesor more. Though this vaccine was tested in humans during the 1950s and 1960s, it was eventually discontinued as the potency of the vaccine was not satisfactory [40]. In the late 1950s, duck embryos became an alternate for chicken embryos in vaccine production, where they developed a duck embryo vaccine (DEV) for rabies, which contained rabies virus in a 10% suspension of whole embryos, inactivated using β-propiolactone. This vaccine was extensively used in humans in the USA until the 1980s. These vaccines were later discontinued due to the adverse reactions and poor antigenicity [41]. The successful adaptation of rabies virus to embryos offered the hope that substitutions could be made for the brain tissue vaccine [42]. Although different strategic approaches could slightly improve the quality of these vaccines, issues regarding their safety, efficacy, and immunogenicity were not fully approved. Consequently, these vaccines were suspended from use in many areas across the globe [30], and research concentrated towards cell substrates for propagating the rabies virus gave rise to the era of cell culture vaccines. 

## 7. Second Generation Vaccines: Cell Culture Vaccines

The creation of cell culture systems for virus propagation has led to a new paradigm for rabies vaccine development, which has resulted in the second-generation cell culture vaccines. The cell culture system is a popular method for producing viral vaccines because it has a number of advantages over nerve tissue vaccines and egg-based systems. It delivers an established safety and efficacy profile, as well as a reduced lead time and higher process flexibility [43]. In 1930, the cultivation of the rabies virus in primary explants of chick embryo brain cells was accomplished, and the virus was further maintained up to five serial passages [44]. Later, research has been focused on the propagation of fixed RABV in mouse embryo brain tissues [45]. In 1942, Plotz and Reagan were successful in the first direct in vitro isolation and cultivation of street rabies virus from the brain of rabid cases in primary explants of chick embryo cells [46]. Later, the concept of rabies virus cultivation in non-neuronal tissue was suggested in 1958, which resulted in the first tissue culture rabies vaccine from primary hamster kidney cells using fixed rabies virus (sourced from rabies infected mouse brain) and street rabies virus (isolated from salivary glands of a rabid dog) [47]. Kissling was able to serially propagate the fixed virus through fifteen cell culture passages and also maintained the street virus successfully through four passages. Following this, in 1960, Fenje achieved the first adaptation of a rabies virus strain using cell cultures for possible use in vaccine production using the Street Alabama Dufferin (SAD) strain propagated in a mouse brain and adapted in primary hamster kidney cell cultures and a mouse brain via alternate passages [48].

Later, in 1963, the first experimental cell culture rabies vaccine, the primary hamster kidney cell vaccine (PHKCV), was prepared by Kissling and Reese, in primary hamster kidney cells seeded with the previously adapted Challenge Virus Standard (CVS) strain of a fixed rabies virus [49]. In 1968, PHKCV, developed using the fixed strain CL-60 (derivative of Street Alabama Dufferin [SAD] rabies virus), was licensed in Canada [31]. Later, in 1971, PHKCV was produced using the Vnukovo-32 strain in the former Soviet Union. After Kissling, scientists have started employing different cell culture systems for the propagation of various strains of rabies virus for vaccine development. In 1964, Abelseth developed an attenuated rabies vaccine for domestic animals using the SAD strain of rabies virus propagated in primary pig kidney cells [50]. Later, in 1965, Kondo exploited the susceptibility of primary chick embryo cell cultures to propagate an egg-embryo-adapted Flury-HEP strain of rabies in developing an inactivated rabies vaccine for human use [51]. In 1969, Wiktor used the BHK-21 cell line, derived from baby hamster kidney cells, for the production of a purified concentrated rabies vaccine [52]. A fetal bovine kidney cell rabies vaccine utilizing the Pasteur virus (PV) and a canine kidney cell rabies vaccine using the Pitman- Moore strain (PM) were both created in 1974 and 1978, respectively. Each of them were licensed for their use in the Netherlands [31].

Slowly, the production of high-quality rabies vaccines has been made feasible through modern cell cultivation techniques using diploid cell strains for vaccine production. This was followed by the growth of fixed RABV in a human diploid cell strain HDCS “WI 38”, which was developed in 1961 by Hayflick and Moorhead [53]. Later in the mid-1970s, WI-38 cell lines were switched over to MRC-5 cell lines in developing a licensed human diploid cell vaccine (HDCV) [54]. HDCV was the first purified, concentrated, and lyophilized rabies vaccine without any adjuvant. In addition, HDCV was reported with much fewer adverse effects. Hence, the WHO has recommended rabies HDCV as a gold standard reference vaccine [30], but human diploid cell vaccines have lower virus yields and are not cost effective enough to be affordable in many developing countries. This necessitated the development of alternative vaccines that are equally effective as human diploid cell vaccines, which led to the development of a purified duck embryo cell vaccine and purified chick embryo cell vaccine.

In 1971, complete duck embryo-derived tissue vaccination, improved using an advanced purification protocol, facilitated the production of a purified duck embryo cell vaccine (PDECV) using the CVS strain [55]. In 1985, the Swiss Serum and Vaccine Institute approved a PDECV vaccine utilizing the Pitman-Moore strain. Further, PDECV is registered in certain European and Asian countries; however, it was not authorized in the USA. These vaccines were demonstrated to be better than DEV as they were completely devoid of egg proteins and myelin basic protein, which was a vital source for allergic encephalomyelitis. Later, in 1972, using the Flury HEP virus, the purified chick embryo cell vaccine (PCECV) was developed. Later with the help of the Flury LEP virus, another PCECV was developed and inactivated, used as vaccines for immunizing dogs for several years. The second PCECV for humans [56] was developed by adapting the Flury LEP strain in chicken embryo fibroblast (CEF) cells and was approved inEurope in 1984. In USA, another PCECV was produced using the LEP-c25 virus andreceived a license in 1997 [57]. Presently, PCECV is one of the most commonly used human rabies vaccines [30]. Due to the comparable immunogenicity and tolerability, purified duck embryo cell vaccines (PDECVs) or purified chicken embryo cell vaccines (PCECVs) have become an effective alternative option for human diploid cell vaccines in human rabies prevention in many parts of the world [58]. 

However, the primary culture system has an inherent constraint with cell divisions. Diploid cell strains, such as WI-38, MRC-5, and FRhL-2, have a limited lifespan of about 50 serial passages, following which, the cells become senescent. Despite their ability to reproduce, it is technically challenging to adapt to large-scale commercial culture for the production of vaccines. This has led towards the use of continuous cell lines for vaccine production. In 1962, a Vero cell line was produced from African green monkey kidney cells. The production of higher virus titers in comparison to those with primary culture cells, the simple scaling up the cell culturing system, and the long history of use in vaccines without raising any safety concerns have made Vero cell lines attractive in the propagation of various viruses. In the early 1980s, rabies virus was propagated in Vero cells for vaccine production [59,60]. The replication of various lyssaviruses, including HEP, CVS, Mokola virus, Duvenhage bat virus, and Lagos bat virus, has been supported by Vero cells. The benefits of Vero cells have considerably lowered the cost of producing the rabies vaccine and made rabies vaccinations accessible for the majority of developing countries. The further introduction of the purified Vero cell-derived rabies vaccine (PVRV) into clinical practice remains crucial for rabies prevention [43]. In 1985, the purified Vero cell rabies vaccine (PVRV) was granted a license in Europe [31]. Moreover, the WHO proposedthe replacement of nerve tissue vaccines with the more effectual, safer vaccines developed through cell culture [20]. Presently, PVRV is widely available and commonly used across the globe. As a step forward in improvisation of the vaccine production process, an enhanced serum-free, PVRV-Next Generation (PVRV-NG) vaccine was produced from the inactivated PM strain of RABV [61]. The immunogenicity and the safety profiles of this next-generation vaccine made it a new alternative for rabies prophylaxis [62].

## 8. Current Developments in Rabies Vaccines 

Despite the fact that numerous anti-rabies vaccinations have been created over the years to protect both humans and animals against rabies, the safety profile and immunogenicity of the vaccine candidate remained of utmost importance. Yet, the essential concepts of vaccine development, such as attenuation and inactivation, have consistently remained the pillars in ongoing rabies vaccine innovations. The continued efforts of rabies vaccine research has largely focused on improving the immunogenicity and safety of the vaccine candidate, which led to the raising of modified or inactivated vaccines.

### 8.1. Modified Live Vaccine (MLV)

Modified live vaccines are usually produced from the naturally occurring virus by altering their pathogenic profiles, so that it mounts a powerful immune response without causing clinical illness. Most scientists modified the virus through serial passage in different cells to ensure the safety of the prospective vaccines, which has led to the development of an attenuated live vaccine for various diseases, including rabies. The original SAD (Street Alabama Dufferin) strain of rabies virus is the source of all attenuated vaccines currently in use, which have undergone varying degrees of attenuation through several passages in cell cultures. Most of the modified live vaccines were attenuated through sequential invitro selection with cloned baby hamster kidney cells or via serial passages in mice in vivo.

In some Asian countries, The Flury strain, a chicken embryo-origin MLV vaccine has been created and administered to animals. Simultaneously, MLV vaccines have used the Street-Alabama-Dufferin (SAD) strain, produced using hamster kidney cells [63] for further use. To create a vaccine that is superior to the Flury-LEP vaccination in terms of quality, in 1974, Canada introduced the attenuated live vaccination strain Evelyn-Rokitnicki-Abelseth (ERA), as the Flury low-egg passage vaccine, which had negative consequences because of tissue debris in the vaccines. Eventually the Flury low-egg passage vaccine was replaced with the ERA strain by the Korean Veterinary Authority in the late 1970s. The domestic animals, including dogs, lambs, goats, and cats, vaccinated with the ERA live attenuated vaccine via an intramuscular route did not show any clinical indications, and the vaccination strain was not recovered from their salivary glands or brains, but it still elicited a robust immune response and elevated viral neutralization antibody (VNA) titers. Unfortunately, nearly 50% of the dogs who received the intracerebral injection of the vaccination experienced significant clinical symptoms, such as anorexia, fever, a severe tremor, paresis, and paralysis [64]. This remains a major drawback of the modified live vaccines due to their residual virulence. Moreover MLV is more sensitive to temperature fluctuations and accidental self-inoculation with the MLV rabies vaccine is quite risky for the vaccinator [64]. Although MLV is more immunogenic and dogs can safely and effectively receive modified live rabies vaccine strains (ERA, Flury, and SAD), eventually, the WHO discontinued recommending MLV rabies vaccines for parenteral injection in 2004.

### 8.2. Inactivated Rabies Vaccine

The history of inactivated vaccines started about a century ago, where the inactivated nerve tissue vaccines developed against rabies were used in certain African and Asian countries. Most of the traditional rabies vaccines use complete, inactivated viruses with the same antigenic properties as wild-type viruses. It has been demonstrated that immunization with entire inactivated virus causes the development of virus-neutralizing antibodies through the activation of helper and cytotoxic T cells and defence against a deadly intracerebral rabies virus challenge. 

Across the globe, the inactivated rabies vaccine has been produced using various rabies virus strains, including CVS 11, Pittman-Moore-NIL2, RC-HL, produced from the Nishigahara strain, and Pasteur virus strains [65]. Currently, the authorized rabies vaccines for human use are based on inactivated purified rabies virus propagated in cell culture or in embryonated duck or chicken egg systems [20]. Usually the rabies vaccine strains are inactivated using beta propiolactone (BPL), ultraviolet light, acetylethylamine, or binary ethylenimine (BEI). The BPL is the most commonly used inactivating agent; however, it is expensive and unstable at 37 °C. Due to their potential to distort the structure of the antigenic sites, phenol and formaldehyde are no longer advised for the inactivation of viruses. In contrast, BEI is less dangerous to handle and offers the benefits of good stability, low cost, and ease of preparation. However, lower immunogenicity, expensiveness, and the requirement ofmultiple vaccination regimens in pre- and particularly post-exposure immunization were the fundamental drawbacks of inactivated vaccines [66]. Hence, to enhance the immunological response, the antigen adjuvants were included in inactivated vaccine. This has led to the development of adjuvanted vaccines. 

### 8.3. Adjuvanted Rabies Vaccines

Adjuvants are substances that increase or modulate the immune response to a vaccine by enhancing inflammatory responses that are essential for the antigen-drivenstimulation of naive B and T cells. Always, adjuvants have drawn much interest because of the use of inactivated vaccines, subunit and synthetic vaccines, which are basically weakimmunogens and adjuvants used as a complement to boost immunogenicity and produce more potent vaccines due of their immune-amplification capabilities [67]. Adjuvants, such asaluminum hydroxide, aluminum phosphate, and saponin, are often used adjuvants [68]. However, the majority of adjuvants used today are aluminum salts. Despite the fact that alum was the first adjuvant approved for use in humans, it is claimed that alum delays the early production of antibodies and is not equally effective for inducing cellular immunity [69]. The adverse side effects, toxicity, and restricted adjuvanticity of certain formulations to few antigens are the major drawbacks with adjuvants, which has necessitated the progress towards the development of synthetic derivatives, suchmuramyl dipeptide, liposomes, QS21, monophosphoryllipidA, MF-59, and immunostimulating complexes (ISCOMS), as alternative adjuvants [67]. In the resent years, much research has been focused on having alternate adjuvants to improve the immunogenicity and effectiveness of the inactivated rabies virus vaccines, and a wide range of other compounds, including Isatisindigotica root polysaccharides, CpG oligo deoxynucleotides, monophosphoryl-lipid A (MPLA), β-glucans, *Staphylococcus aureus*-derived hyaluronic acid (HA), and bacillus Calmette-Guérin purified protein derivative (PPD), were studied. These candidate adjuvants could definitely be a promising platform for the development ofnovel adjuvanted rabies vaccines in the future with enhancedimmune-boosting potential and a reduced risk of adverse health effects [70,71,72,73,74,75].

## 9. Next-Generation Vaccines

The risk posed bythe residual virulence of modified live vaccines and the low immunogenicity and requirement of huge amounts of antigen and repeated doses to elicit a protective immune response by the inactivated vaccines warrants the need for more safe and affordable rabies vaccines. Presently, the advent of technologies, such asrecombinant DNA technology and reverse genetics, have given an in-depth view of the rabies viral genome and facilitated genetic manipulations, which in turn, holds much promise for more potent and safer vaccines, with lower costs and improved stability and immunogenicity. This has led to the era of next generation-vaccines targeting recombinant rabies virus strains or individual recombinant rabies antigenic glycoprotein (G protein), which were helpful in overcoming the disadvantages of the live attenuated vaccines [64].

### 9.1. Genetically Modified Vaccines

Genetically modified viruses are generated through genetic modification of the viral genome through the directed insertion, deletion, artificial synthesis, or change in nucleotide sequences through biotechnological methods and still retain the infection capabilities.Most rabies vaccines include attenuation, weakening, or inactivation of the viruses in some manner so that their virulent characteristics are rendered ineffective. The genetic focus on the genome of the virus has confirmed that glycoprotein (G) is most related to RABV pathogenicity and has discovered certain amino acid sites related to viral pathogenicity [76].Thus, further genetic manipulations of the parent rabies virus strain in terms of creating site-specific mutations in these amino acids or the insertion of modified glycoprotein will abolish residual pathogenicity, eliminate potential reversion to virulence, reduce a potential back-mutation to the original amino acid, and enhance safety in the resultant mutant, which in turn, will be a promising option to generate highly attenuated rabies vaccines [23].

Especially, these vaccines are crucial for the mass immunization of stray dogs in developing countries or even the wild animal habitats of developed countries, and many mutant vaccination strains have been developed as a result of various research efforts. A genetically modified ERA vaccine strain wild-type (rERA) with an arginine-to-glutamic acid mutation at residue 333 of RVG (G333E) was developed. Through intramuscular (IM) inoculation in dogs and also oral immunization in mice, this live rERAG333E could induced potent and long-lasting RVNA that washigher than that with the ERA strain wild-type (rERA) [18]. Furthermore, using reverse genetics, the complete genome was mutated at the amino acid position 333 of the RVG gene of rRABV, and helper plasmids were rescued in BHK/T7–9 cells, resulting in construction of the ERAG3G strain, which with further oral immunization in mice, displayed total defence against pathogenic RABV [19]. Another novel strain, ERAGS, a rRAVB with site-specific mutations at positions 194 and 333 of the RVG, was found to be non-pathogenic, extremely safe, and highly effective against highly pathogenic RABV in IM-vaccinated mice [77]. A novel and a highly attenuated double glycoprotein rabies virus construct, SPBN GASGAS, was generated from the rabies strain SAD L16 through mutations at amino acid positions 194 and 333 and with an additional identically altered glycoprotein gene. The resultant mutant has an improved safety profile through a reduction in the potential risk of reversion to virulence and enhancement of apoptosis. Recently, SPBN GASGAS has taken a special place as a potent rabies vaccine candidate for oral vaccinations to mitigate rabies in free-roaming dogs of certain developing countries, such as Thailand, Haiti, Namibia, and Morocco [23].

### 9.2. Recombinant Rabies Vaccines

Owing to safety concerns and improved immunogenicity, further recombinant vaccine variations were created by editing the rabies virus genome to encode two or more copies of the glycoprotein or using strategies to clone and express only the rabies virus glycoprotein (RAVG). A recombinant rabies vaccine construct encoding two copies of the glycoprotein gene provided better protection in mice and conferred protection in dogs against a virulent challenge with the Challenge virus standard-11 (CVS-11) strain [78,79]. The studies also revealed that RABV containing two copies of the G gene has increased expression of the G gene, which considerably improved the efficacy of the vaccine with enhanced immunogenicity with higher levels of viral neutralizing antibody production and decreased pathogenicity [80,81]. The elevated G gene expression was associated with enhanced apoptosis, which contributes to the induction of the strong upregulation of genes related to host immune responses observed in neurons infected byattenuated RABV strains [80,82]. Thus, these recombinant RABV strains may be a promising inactivated vaccine candidate in the future.

### 9.3. Nucleic Acid-Based Rabies Vaccines

Genetic material from a disease-causing virus is used in nucleic acid vaccines to produce a protective immune response against infectious agents. Vaccines made from nucleic acids have the potential to be cost-effective, safe, and efficient. Moreover, the immune responses brought on by nucleic acid vaccines solely focus on the selected pathogen antigen. Nucleic acids vaccines can be of DNA (as plasmids) or RNA [as messenger RNA (mRNA)] and show remarkable potential for addressing a variety diseases [83]. Thisis based on cloning the DNA into a delivery plasmid or through thedirect inoculation of messenger RNA (mRNA) to express antigens inthe host cells. Further, the host machineries will aid in endogenous protein synthesis, which mimics a natural infection triggering both cellular and humoral responses against the expressed antigen.

#### 9.3.1. Rabies DNA Vaccines

Generally, adequate levels of virus-neutralizing antibodies are directed against the rabies virus glycoprotein, which in turn, correlates with rabies protection. The glycoprotein gene can be easily cloned into the proper expression vectors using the techniques of recombinant DNA technology, which facilitates the efficient in vivo expression of glycoproteins and resulted in the production of rabies DNA vaccines. Since 1994, DNA vaccines have been suggested as a more affordable and effective method for rabies prophylaxis, and theirviability has been shown in a variety of animal models, including companion animals [84]. The glycoprotein sequences of the Pasteur virus (PV), Challenge virus standard (CVS), Evelyn Rokitnicki-Abelseth (ERA), or street viral isolates were employed to assess DNA rabies vaccines. The RV glycoprotein was expressed in various expression vectors, including pSG5, pCIneo, pVR105, DNAVACC and was successful in conferring a specific immune response in the experimental animals [21,85,86,87]. Though the technology has been shown to be effective, still its widespread use, especially in post-exposure prophylaxis against rabies, is prevented by the slower and weaker immune responses.However, the co-administration of a glycoprotein-encoding plasmid with chemical adjuvants or co-delivery of cytokine genes and innovative delivery techniques, including DNA injection followed by electroporation and co-administration with a traditional inactivated rabies vaccine, were efficient in enhancing the immune responses and increased their potential for rabies prevention and management [84,88]. Moreover, the current developments in vector design and delivery systems also hold promise for improving the effectiveness of rabies DNA vaccines [89,90]. However, the risks associated with the integration of plasmid DNA into a host chromosome, development of tolerance against the plasmid DNA vector, antigens, and auto-immunity need to be crucially addressed before they are developed into a widely used vaccine.

#### 9.3.2. Rabies RNA Vaccines

RNA vaccines are the simplest nucleic acid vaccines and emerged as a potentialmorepromising alternative platform for vaccine development. RNA vaccines have a number of advantages over DNA vaccines. RNA vaccines are directly translated in the cytoplasm, eliminating the need for transport into the nucleus and resulting in fast antigen expression with no possible risk of integration with the host genome. It is a known fact that RNA is a powerful adjuvant that increases the host immune response by communicating through pattern recognition receptors, such asthe Toll-like receptors or the retinoic acid-inducible gene I-like receptors [91]. The ability of messenger RNA (mRNA) to self-replicate has led to the development of two main types of RNA vaccines at the moment: traditional non-amplifying mRNA vaccines and self-amplifying mRNA vaccines (also known as replicons) derived from an RNA virus vector, such asthe Semliki Forest virus (SFV),whichmaintains replicative activity [92]. Research has revealed that the traditional non-amplifying mRNA-based rabies vaccines encoding the RABV G gene stimulate potent VNAs and antigen-specific CD4^+^ and CD8^+^ T lymphocytes in inoculated mice and protect them from lethal intracerebral challenge infection [93]. Similarly, mRNA-based rabies vaccines were found to be immunogenic in domestic pigs with effectual VNA levels [93]. A self-amplifying mRNA-based rabies vaccine with a recombinant SFV containing an RNA-coding RABV G protein (SFV-RVGP) caused high expression levels of functionally trimeric RABV G protein, evoked comparable levels of antibodies to thoseof a commercial rabies vaccine, and was more effective than the protein vaccines in generating a cellular immune response [22]. However, a major challenge is the instability of RNA vaccines due to RNase-mediated degradation or the resulting electrostatic repulsion due to the interaction of the negativelycharged mRNA molecules with the negativelycharged cell membrane results, leading to the transient expression of antigens after RNA delivery, which needs critical attention.

### 9.4. Protein Subunit and Peptide Vaccines for Rabies

Protein vaccines stimulate the immune system by using peptides or proteins as antigens. Protein antigens can be produced naturally from the pathogen that causes the infectious disease, either in their entire form or as derived split-products. Protein subunit vaccines can also be created as heterologous proteins in recombinant systems, including recombinant bacteria, yeast, insect cells, or mammalian cells, as an alternative to natural sources [94]. The RABV G protein is the main determinant of viral pathogenicity and is also the main protective antigen responsible for inducing protective immunity against rabies. Hence, the effectiveness of G protein (purified from infected cells) has been determinein various systems and used as aform of protein vaccines for rabies protection. However, these were not found to be effective. Yeast-derived protein failed to elicit a protective immunological response in mice, which was most likely due to poor folding of the end product [95]. It was shown that baculovirus-derived G protein produced in insect cells was immunogenic [96], but the considerable purification needed would probably make this method uneconomical. Currently, the use of geneticallyaltered plants to produce vaccines is gaining much significance. Researchers havealso been employing plants, such astomato, maize, cantaloupe melon, and tobacco to express the RABV G protein in order to create edible vaccines [97,98]. This vaccine could generate a detectable VNA response and protection against challenges in mice upon ingestion. Although these types of vaccines could potentially be produced with alow cost, there areseveral obvious drawbacks, including immunogenicity, the vaccine’s stability in fruit, its degradation in the stomach, and the intestinal immune response. Consequently, despite the fact that edible vaccines are a desirable option, much more research is required before they can be trusted to replace rabies shots.

Apart from protein vaccines, a number of antigenic epitopes of the RABV G protein were identified, and synthetic peptides mimicking these G protein epitopes have been produced and used as vaccines. Animals inoculated with a synthetic peptide encapsulating the rabies virus’s G5 antigenic region produced antibodies with strong avidity but low neutralization potential [99]. Peptide mimotopes of rabies virus glycoprotein (RABVG site III mimotope (C-KRDSTW-C) werefound to be more immunogenic in mice and have provided a new concept of rabies vaccinesfor thefuture [100]. However, vaccines made of proteins or peptides are moderately immunogenic; the 65kDa viral glycoprotein is synthesized, forms trimers, and is moderately N-glycosylated at one of three possible locations. The rabies virus glycoprotein must fold correctly into its native trimeric form in order to produce antibodies that can neutralize the virus, but this is still a challenge for protein vaccines [37].

### 9.5. Parenteral Viral Vector Rabies Vaccines

Viral vector vaccines were developed by cloning the antigen of interest into a heterologous virus that will serve as a carrier molecule to transfer the foreign gene of interest within the host cells, thus leading to its subsequent expression [101]. The first viral vector expressing a foreign gene was created from the Simian Virus 40 in 1972; since then, different viruses, including adenoviruses, poxviruses, herpesviruses, vesicular stomatitis virus, and lentiviruses, have been genetically modified to render them non-pathogenic and engineered into vaccine vectors that can potentially stimulate the immune system against the proteins generated from the encoded transgenes. The antigen-specific cellular immune responses and strong and long-lasting antibody levels elicited by the viral vector with their inherent adjuvanticity has made them more potent candidates in rabies vaccine development.

Several researchers targeted different viral vectors for the expression of rabies viral glycoprotein. The recombinant adeno-associated viruses (AAVs) with RVG vaccinations enhanced the protective levels of Rabies Virus Neutralizing Antibodies (RVNAs), with single intramuscular vaccination [102]. Several species of Adenoviruses (Ads) have been extensively used as vaccine vectors in vaccine development as they bring forth strong humoral and cellular immune responses [103,104]. Especially, simian Ad vectors encoding rabies virus glycoprotein were tested for rabies vaccination. A chimpanzee adeno vector-based rabies virus glycoprotein clone AdC68-rab. GP enhanced RVNA titers in mice to levels beyond suggested standards four weeks after vaccination through intramuscular injection [105]. A single intramuscular injection of ChAd155-RG, which is based on a group C chimpanzee Ad vector, induced high RVNA titers and more potent cellular immunity in mice, as well as very rapid, sustained, robust, and durable RVNAs in rabbits and protective levels of RVNAs lasting up to 48 weeks in macaques. These effects are comparable to those induced by one, two, or even three doses of the Rabipur vaccine [106]. The ChAdOx2 RabG with chimpanzee Ad serotype 68 (AdC68) and the human Ad serotype 5 (AdHu5) were found to be more immunogenic and elicited high levels of RVNAs only witha single IM dose in vivo in comparison to those with other Ad serotypes. The protective immune responses generated by theCanine adenovirus 2 rabies virus glycoprotein (CAV2-RVG) through intramuscular, intranasal, and oral immunization in mice and dogs were quite appreciable [107,108]. Furthermore, other viral vectors, including Parapoxvirus Orf virus (ORFV) [109], vesicular stomatitis virus (VSV) [110,111], raccoon poxvirus (RCN) [112], single-cycle flavivirus [113], Newcastle disease virus (NDV) [114], and bovine herpes virus type I (BHV-1) [115] expressing rabies virus glycoprotein, effectively induce RVNAs and remain as potent vaccine candidates. Still, a common obstacle with the viral vectors is the host’s preexisting immunity to certain vectors that could minimize the efficacy of the vaccine. However, this can be circumvented with the use of alternative vectors that have low seroprevalence in the host. Moreover, the development of viral vectored rabies vaccines has taken its step forward as oral rabies vaccine candidates and foundational tools to reduce the toll of rabies virus in wildlife species across the globe.

## 10. Oral Rabies Vaccines (ORVs)

Animal vaccination is an effective way of protecting public health through the reduction of zoonotic pathogens. The long history of rabies vaccine development has landed with many potent vaccines for parenteral vaccination strategies. However, the vaccination of wildlife or free-roaming domestic animals, such asstray dogs, through parenteral vaccination remains a challenge. Hence the need for an alternative vaccination strategy for the eradication of rabies has led to the development of oral rabies vaccines (ORVs). The oral vaccines are given orally using a bait designed with the vaccine suspension-sealed sachet encased in a palatable bait material, such as animal intestine, chicken head, egg, and dog food-based materials, depending on the target species. Upon consumption, the chewing motion causes the sachet to be perforated, releasing the vaccination suspension into the mouth. Here, the vaccine is mostly absorbed by the palatine tonsils where, following minimal replication at the site of entry, it triggers a protective immune response [116]. Currently, two types of commercial licensed ORVs are available for immunizing against wildlife rabies. They are modified live vaccines (MLVs) and vector-based vaccines (VBVs). In modified live vaccines, the rabies virus is modified to attenuate its virulence, but still with theability to trigger the immune system for the production of antibodies [117]. In the vector-based vaccines, the genes coding for the antigenic glycoprotein areinserted into a viral vector, which can induce an immune response by expressing the inserted rabies virus glycoprotein upon immunization. Since 1978, oral rabies vaccines (ORVs) have been used successfully to control wild animal rabies across Europe and the USA.

### 10.1. Modified Live Rabies Virus Oral Vaccines

The development of modified live rabies virus oral vaccines started way back in 1935, derived from the Street Alabama Dufferin (SAD) rabies virus strain. This parental SAD strain was isolated from the salivary glands of a rabid dog in the USA. Subsequent serial passages of the SAD strain in unnatural hosts, such asmice, chick embryos, and non-neural cell lines, includinghamster kidney and pig kidney cells, with further heat stabilization, has resulted in thehighly attenuated SAD-Bern, Evelyn Rokitnicki Abelseth (ERA), and SAD-B19 strains. These strains constitute the first-generation ORVs, which werethe keystone for controlling wildlife rabies in Europe and havebeen widely used ORVs across the globe [118]. Further, to improve the safety profile of the first-generation ORVs, selection mutations were induced in the rabies virus SAD strain using monoclonal antibodies, resulting in second-generation ORVs, SAD VA1, SAG1, or SAG2 strains [119]. A double avirulent mutant SAG2 strain was isolated from the SAD-Bern strain of rabies virus in1990 throughtwo successive mutations affecting the amino acid residue 333 of the glycoprotein. The SAG2 strain was shown to be an improved version of SAG1 due to its safety and genetic stability [120,121]. Though these vaccines were successful inkeeping wildlife rabies at reduced levels, theeffective control of rabies is still required among the large free-roaming dog populations in most rabies-endemic countries. This necessitates a safer vaccine candidate that can complement parenteral vaccines in mass dog vaccination campaigns, which remains crucial [23]. Hence, modern technologies of reverse genetics have facilitated site-directed mutagenesis, and targeting specific changes at selected locations in the rabies virus genome has led to the development of third-generation MLVs. Presently, SPBN GAS GAS and ERA G333 are the two 3rd generation vaccines currently tested for use in canids [122,123]. Although both vaccine strains are from different parent strains, still, they have alike mutations at amino acid residue 333 of the G-protein [118]. The safety and immunogenicity of the existing MLVs have been further improved by these site-specific deletions and insertions. However, a major concern with MLVs is the possibility of the vaccine virus undergoingrandom mutations, which wouldprobably revert them to virulent forms causing rabies [124]. Intracranial inoculation of first generation MLVs was reported to cause rabies in immunosuppressed mice [125]. In addition, reports revealed 11 cases of vaccine-associated rabies in immune-suppressed foxes and non-target species in Europe following vaccination with first generation MLVs, accounting for 1 in 48 million bait doses distributed [119]. In contrast, no field cases were reported in this regard with second- and third-generation vaccines.

The success of field trails in generating immune responses in free-roaming dogs whenusing SPBN GASGAS was appreciable in certain developing countries. Using oral bait with SPBN GASGAS supplied in Thailand for 2444 dogs, which were not accessible for parenteral vaccination, about 65.6% of the roaming dogs at these locations were effectively immunized orally [126]. An evaluation of immune responses in dogs to oral rabies vaccine through mass dog vaccination under field conditions in Haiti revealed immune responses in 78% of dogs that consumed the bait [127]. By monitoring the immunological response in terms of seroconversion for up to 56 days after vaccination in local free-roaming dogs from Namibia, the immunogenicity of the highly attenuated vaccine strain SPBN GASGAS was successful in protecting approximately 79% of the dogs through oral vaccination [128]. These investigations confirmed the immunogenicity of the vaccine strain SPBN GASGAS and the ability of ORV to reduce dog-mediated rabies in developing countries. Moreover, the viability of using oral bait distribution to reach stray dogs in Goa, India, for rabies vaccination was experimentally demonstrated, and thatstudy supported the contention that ORV will be a new approach promising a significant increase in dog vaccination coverage. In addition, ithighlights the use of ORVs in India as a complement to parenteral mass dog vaccination programs due to its feasibility and ability toaccess thefree-roaming dog population in the country [129].

### 10.2. Viral Vector-Based Oral Rabies Vaccines

The development of vector-based vaccines helped to overcome the risks associated with MLVs. These viral vector-based rabies vaccines are more promising for both veterinary and human use [113,130,131]. Several research efforts have resulted in effective viral vector-based ORVs. A recombinant vaccinia virus expressing the glycoprotein of the rabies virus was tested as an oral rabies vaccine in bait and was shown to induce protective immunity in several wild animals [132]. Several species of adenovirus have been extensively used as vaccine vectors in vaccine development as they bring forth a strong immune response [103,104]. The development of oral vaccine baits based on replication-competent human adenovirus type 5 (AdHu5) expressing the rabies glycoprotein (ONRAB) was encouraging in controlling wildlife rabies in North America [133]. The efficacy and safety of 1st generation replication-defective adenoviral vectors were improved when compared to those with a 2nd generation replication-competent adenoviral vector [130]. Several reports revealed that the rabies virus glycoprotein expressed in E1-deleted AdHu5 vectors was found to be more promising in rodents and canines [134,135]. However, the pre-existing immunity with detectable virus neutralizing antibodies (VNAs) against AdHu5 in 45–90% of the human population was found to dampen the immune responses elicited by the AdHu5 vector [130]. In order to overcome these issues, presently, rare serotypes of human adenovirus or non-human species adenoviruses, such as chimpanzees and canines, have been used for vector construction and for vaccine development [104]. Intranasal or oral routes of administration of a recombinant chimpanzee adenovirus serotype 68-based vaccine (AdC68) expressing the rabies virus glycoprotein elicited effective VNAs in newborn mice [136]. This vaccine efficacy was found to be encouraging in young, pre-exposed individuals through an oral route. In addition, chimpanzee adenoviral vector-based rabies vaccines were shown to protect dogs from a lethal rabies virus challenge [137]. Studies recorded a protective immune response with Canine adenovirus 2 rabies virus glycoprotein (CAV2-RVG) through oral immunization in mice and dogs [107,108].

Currently, two commercially licensed VBVs are available for controlling wildlife rabies viz., RABORAL V-RG, which uses recombinant vaccinia virus vector [24], and ONRAB, with a recombinant human adenovirus vector [25]. One of the major issues with the use of VBVs is the probable infection caused by the vector virus itself. Studies have highlighted that exposure to recombinant vaccinia virus vector vaccines could cause severe skin inflammation in humans and complications in pregnant and immunocompromised individuals in the USA [138], but such public concerns have not been reported with Adenoviral vectors. Moreover, the potential interference withpre-existing immunity against the vector is amajor concern with vector vaccines, as it prevents the generation of an adequate immune response against rabies [25]. However, present viral vectors have been engineered in such a way to evade pre-existing immunityand to make them more of a potential delivery system [139]. The success stories of ORVs in the effective control of wildlife rabies and theirpotential to trigger immunity in experimental studies highlight the efficiency of ORVs as an alternative strategic vaccination to complement parenteral vaccination in controlling rabies in developing countries.

The details of the current research in the development of different vaccines and advantages and disadvantages of different vaccines are summarized in Table 1 and Table 2.

## 11. Intradermal Rabies Vaccination

The ability of a vaccine to stimulate the immune response is also dependent on the route of inoculation of the vaccine. Intradermal vaccination delivers antigens into the space between the epidermis and the dermis. This space is an anatomically favorable site for immune stimulation, as the dermis and epidermis layers of the skin are abundant sources of antigen-presenting cells, such as Langerhans cells, dermal dendritic cells, and dermal macrophages, which are known to take part in vaccine-induced immune responses, and skin is seen as a desirable vaccination target. The dermis’ extensive network of blood capillaries and lymphatic arteries also makes it easier for leukocytes and dendritic cells to travel from the skin to the secondary lymphoid organs. Most rabies vaccinations administer an increased amount of antigens via the intramuscular (IM) route. Hence, its usage is restricted in many under-developed nations due to the price of these antigens and occasionally their scarcity. Reduced dosages (usually 10% or 20% of the standard dose of antigen) administered via the intradermal (ID) route areable to elicit immunological reactions similar to those brought on by the standard dose administered via the IM route [158].

In the United States, pre-exposure immunization with the ID method of rabies vaccines was first authorized in 1986, which was followed by the development of a cost-effective multi-site intradermal (ID) immunization approach. The intradermal administration of contemporary rabies vaccines for post-exposure prophylaxis was advised by the WHO Expert Committee in 1991 [26]. According to a WHO review, the cell culture rabies vaccine (with a potency of >2.5 IU per intramuscular dose) given via the ID route for post-exposure prophylaxis (PEP) or PrEP has efficacy on par with or higher than that of the same vaccine given via the IM route in humans [159]. Similarly, the effectiveness of rabies pre-exposure prophylactic vaccines given by various methods in cattle revealed appreciable levels of antibodies that neutralize the rabies virus (RVNA) through the ID route [157]. A combined treatment protocol for rabies wound infiltration with eRIG and a 5-day (0, 3, 7, 14, and 28) PEP regimen through the intradermal route was life-saving in humans, cattle, and dogs and theID route was proven to be more economical due to its dose-sparing effect [155]. Similarly, inactivated cell culture rabies vaccine administration viaSC, IM, and ID was found to be safe and immunogenic [156]. These clinical trials provide confidence to promote the ID delivery of rabies vaccines in areas where access to PEP is hampered by expense, and ID should be encouraged as a cost- and dose-saving alternative immunization strategy over intramuscular immunization.

## 12. Immunity to Rabies Virus: Natural Infection vs. Rabies Vaccines

### 12.1. Immunity to Natural Rabies Virus Infection, including Humoral and Cellular Responses

RABV typically infects the nervous system (NS). Yet, in the early stages of infection, virus particles are “injected” into the muscles and skin, and before the virus enters the nervous system, it might cause an immunological reaction in the periphery [160]. Once RABV enters the NS, it is unlikely that it will cause a primary adaptive immune response, as the NS is an immune-privileged location because it lacks professional antigen-presenting cells and lymphoid tissues. However after entering the NS, RABV causes an early innate immune response after infection that is characterized by antiviral, chemo-attractive, and inflammatory responses, all of which involve infected neurons [161]. Due to the insufficient immune efficacy in the NS, RABV uses immunosubversive tactics to evade the host immune response, thus successfully adapting to the host system, and the cell-mediated immune (CMI) response plays a critical role inabolishing viral invasion and inhibiting the entry of RABV into the peripheral nervous system (PNS) at the beginning of the infection. However, the blood–brain barrier is a significant obstacle for the CMI response. Moreover, the expression of HLA-G molecules on neuronal cells viainterferon beta production results in the transformation of naive human T cells into regulatory T cells, and the production of inhibitory cytokines, such asinterleukin 10, results in cellular immunosuppression [162]. In addition, the humoral immune system is a key defense mechanism against the rabies virus. Neutralizing antibodies can block viral infectivity and stop the virus from adhering to neuronal cells in immuneindividuals. According to studies, unvaccinated animals when exposed to RABV are unable to produce the optimum level of neutralizing antibodies, which allows the virus to reach the nervous system, manifestingas the disease [162,163].

### 12.2. Immunity to Rabies Vaccines, including Humoral and Cellular Responses

The most crucial function of rabies vaccination is to induce a persistent antibodyresponse through the activation of CD4^+^ T lymphocytes. Though it is typically believed that cytotoxic T cells are more crucial than antibodies to remove virus infection from tissues, rabies is an exception. Moreover, CD8^+^ T cell activation results in a pathogenic reaction that is clinically linked to paralysis. Based on the high risk of generating a potent, harmful CD8 response in the nervous system, this knowledge should generally prevent the use of live vaccines, such as DNA vaccines or recombinant viruses, as post-exposure immunizations [164]. Yet, due to the effectiveness of live immunization, these new-generation vaccines would still be suitable for pre-exposure vaccination regimens. The best option for maintaining the integrity of the nervous system is the inactivated post-exposure vaccines, which primarily promote B cell activation with the aid of CD4^+^ T cells. Because post-exposure immunizations set up an immune response in peripheral secondary organs, they are likely to provide protection. Activated lymphocytes, CD4^-^ producing plasmocytes, secrete antibodies thatgain entry into the nervous tissue parenchyma and neutralize the virus [165].

## 13. Therapeutic Approaches to Rabies Using Vaccines and Antibodies

### 13.1. Rabies Vaccine-Based Therapy

Thousands of people die from rabies each year, with the majority of those deaths occurring in Asia and Africa. However, rabies can be prevented throughvaccination if post-exposure prophylaxis (PEP) is administered promptly and effectively. Rabies exposure is roughly divided into three categories: I, II, and III, depending on the nature of the probable rabid animal interactions. Any exposures that are found to provide a rabies risk necessitate PEP, which involves vaccination, the thoroughwashing and disinfection of all bite and scratch wounds, local wound infiltration with RIG (for category III alone), and the prompt local treatment of all bite and scratch wounds [166]. The primary objective of PEP is to stop the onset of clinical rabies following exposure. The PEP regimen generally follows the administration of human rabies immune globulin (HRIG) only once, followed by the first vaccination on day 0 of treatment and the following threeimmunizations administered on days 3, 7, and 14, thus ensuring sufficient virus-neutralizing antibodies (VNAs) and conferring protection against rabies. In cases of immunocompromised people, a fifth vaccination on day 28 is advisable and the seroconversion is tested from 7 to 14 days following completion of the PEP regimen. Onlytwo booster doses of the rabies vaccination, administered on days 0 and 3, should be given to patients who have previously received either pre-exposure or post-exposure rabies prophylaxis. People with prior vaccinations should not receive HRIG [166]. As per WHO norms, VNA titers more than 0.5 international units per mL of serum areconsidered adequate for conferring protection in both humans and animals.

### 13.2. Rabies Immune Globulin (RIG)-Based Rabies Therapy

Rabies Immune Globulin (RIG) is either a polyclonal or monoclonal antibody that specifically targets the G protein epitope, which wasfound to be very effective at neutralizing RABV prior to the virus entering the central nervous system (CNS) [167]. This will provide immediate antibodies until the body responds to the vaccine by actively producing antibodies. RIG is available in two varieties, human rabies immunoglobulin (HRIG) and equine rabies immunoglobulin (ERIG), and in general, 20 IU/kg of body weight HRIG and 40 IU/kg for ERIG dosages are used to neutralize the rabies virus [168]. Generally, the area around the wound site should be injected with it, ideally the day of exposure or up to 7 days following the initial dose of vaccine, which helps in the initial neutralization of the virus at the neuromuscular junction. RIG continues to be an essential component of post-exposure prophylaxis (PEP) because it offers passive immunity at a critical stage before the host develops active immunity to the immunization. Unfortunately, these antibodies are limited to crossing the immune-privileged blood–brain barrier (BBB) to counteract virus infection once the virus enters the CNS. However, studies indicate that the effectiveness of the simultaneous administration of RIG and BBB permeability-enhancing agents, such ascytokine MCP-1 or hyper-osmotic solution hypertonic arabinose, through intravenous injections, critically allows antibodies to cross the BBB and enter the central nervous system, neutralizing RABV from the CNS and preventing the development of rabies in mice and rats. Thus, such therapeutic approaches of RIG can serve as a promising option in the treatment of rabies [17,169].

### 13.3. Small Interfering RNA (siRNA)-Based Rabies Therapy

A class of double-stranded RNA molecules of about 21–23 bp, known as siRNA, also referred to as short interfering RNA or silencing RNA, repress the expression of particular genes by degrading mRNA following transcription via the RNAi pathway. The siRNA-based gene silencing of target genes has become an option for antiviral defense against many illnesses and disorders. Research has focused on plasmids encoding siRNA or viral vector-based systems of siRNA targeting specific RABV genes, which were constructed and tested for anti-rabies efficacies [170,171]. Especially, viral vector-based siRNAs often provide significant protection against lethal RABV challenge and effectively suppress the related gene expression and RABV proliferation. Recent research also highlights that siRNAs can serve as potential RIG substitutes for RABV treatment possibilities [172].

## 14. Future Prospects of Rabies Vaccines

The recent advancements in rabies vaccine developments have generatedpromising tools for preventing and eradicating rabies. However, there are certain areas to be focused on in the near future. The current development ofthe newly invented vaccines is quite appreciable; however, the development of novel vaccine delivery systems remains a vital areas of study, which will help to formulate effective strategic measures to eradicate this illness. Failure of the rabies vaccines due to insufficient vaccine potency as a result of exposure to unfavorable temperatures during storage andshipping critically necessitates research on the developmentof potential rabies vaccines that can be stored and transported outsidethe usual cold chain, which will definitely revolutionize vaccine distribution by increasing the effectiveness, efficiency, and affordability ofvaccine delivery. Recent research on sugar-matrix thermostabilization (SMT) technology for viral-vectored vaccines will be a novel platform in the future for eliminating theneed for the cold-chain storage of vaccines. Further exploration for novel adjuvants (cPG, colloidal manganese salt, BAFF, etc.) will definitely be a hopeful option in improving the potency of inactivated anti-rabies vaccines. Application opportunities for the creation of multivalent vaccines arepromising. The development of successful bivalent viral vector-based vaccines against rabies and canine distemper will have potential to control both diseases with a single vaccine candidate, especially in developing countries. Moreover, future vaccine development should focus oncreating a multivalent vaccine with broad-spectrum protection efficacy considering other members of Lyssavirus, which will be very effectual in curtailing this deadliest disease. Although the quality and availability of rabies vaccines and RIGs produced from cell cultures have progressively improved, there has consistently been modest interest in creating antiviral therapy, which warrants research on potential therapeutic interventions for clinical application, through potential antiviral medications. More research on rabies vaccines using small interfering RNA (siRNA) in the natural host or the development bi-specific antibody (BsAb)-based therapies for rabies virus holds more promising options in the therapeutic aspects of rabies. Moreover, improving post-exposure vaccination in animals and redefining the regimen of vaccination forvarious types of vaccines remain the need of the hour. The further development of novel andimproved immunogenic oral bait vaccines in target hosts will be much appreciated in controllingrabies, especially in the free-roaming dog population in developing countries through mass dog vaccination programs.

## 15. Conclusions

The saga in the development of anti-rabies vaccines and the path traversed in the development of rabies vaccines from Pasteur to the modern era of immunization has faced many ups and downs. However, still, these pioneering works have laid a strong foundation for the successful development of vaccines to prevent human deaths and curtail canine rabies at the moment and thus holds much appreciation. Moreover, the roadmap ahead in vaccine development with advanced scientific technologies to manipulate the rabies viral genome and novel vaccine carriers will definitely provideoutstanding developments in vaccine research in the near future. However, the immediate need of countries endemic for dog-mediated rabies is to employ mass dog vaccination using the parenteral vaccines for domestic dogs and accessible community dogs, complemented by ORVs for free-ranging dogs that are difficult to be caught. This combined approach could make the Global Strategic Plan for the global elimination of dog-mediated human rabies deaths by 2030 a reality.

## Figures and Tables

**Table 1 vaccines-11-00756-t001:** Current research status of rabies vaccines.

Vaccines	Vaccine Name	Research Focus	Research Outcomes	References
Modified Live rabies vaccine	ERA- G333 Leu	Arg-to-Leu mutation at G333 using reverse genetics in ERA strain	Increased neutralizing antibody response and protective immunity in 6-week-old mice and increased their survival rate	[140]
	rSAD- K83R	Lys83-to-Arg83 and Pro367-to-Ser367 in the G protein of the RABV SAD strain using site- directed mutagenesis.	Increased the level of RABV-G expression, which also caused more apoptosis in infected cells The K83 mutation increased the expression of MMP-2 and MMP-9 inDCs and increased BBB permeability	[141]
	rGDSH- D255G	Asp255-to-Gly 255 mutation	Reduced pathogenicity andneurotropismof RABV in adult mice	[142]
	CTN181 –3	G276 (Leu-to-Val) and L1496 (Met -to-Trp)in the parental virus strain CTN-1	Raised RVNA levels and seroconversion rates to 100%	[143]
Inactivated rabies Vaccine	- *	Vero Cell Rabies Vaccine Inactivated and stabilized using different inactivating compounds	Increased IgG levels	[144]
	-	Studied the potency of βPL, BEI, and H_2_O_2_ inactivated rabies vaccines	Increased the levels of IgG with BEI and H_2_O_2_ inactivaed vaccines. Significant levels of IFN-γ with BEI inactivaed vaccines and elevation of IL-5 levels with βPL and BEI inactivaed vaccines.	[75]
Adjuvanted rabies vaccine	-	Effect of adjuvanticity β-glucans on inactivated rabies vaccine (Rabisin^®^)	Amplified adaptive immune response	[74]
	-	Effect of 1,3-1,6-glucans on cats’ rabies immunization levels	Promoted the synthesis of RVNA	[145]
	-	Effect of colloidal manganese salts in enhancingrabies vaccination effectiveness in mice, cats, and dogs	Boosted the immunogenicity and protection rate of rabies vaccines by increasing the numbers of mature DCs, Tfh cells, GC B cells, PCs, and RABV-specific ASCs in mouse models	[146]
	RABV- ED51- mBAFF	Studied the immune response of B cell activating factor (BAFF) withrabies virus immune response	Raised the level of particular IgM, IgG, IgG2c/IgG1, and RVNA synthesis	[147]
	LBNSE- U-OMP 19	Investigated the immunogenicity of the recombinant LBNSE-U- OMP19	Increased levels of RABV-neutralizing antibodies and better protection in mice immunized following oral immunization	[148]
Nuclei based acid rabies vaccine	RG SAM (CNE)	Assessed the rabies self-amplifying mRNA vaccine in rats	Increased immune response in rats	[149]
	pVax-G -cons- CD63	Studied the effectiveness of the rabies virus glycoprotein with a consensus amino acid sequence and a lysosome-targeting signal	Increased immune response in mice	[150]
	pVaxF1	Study of vaccination against rabies using a DNA vaccine expressing the G5 linear epitope and the C3d-P28 adjuvant	Increased the production of RVNAs in mice	[151]
Recombinant vaccines	NC8- pSIP409 -dRVG	As a new oral rabies vaccine, recombinant Lactobacillus plantarum NC8 delivers one or two copies of G protein linked with a DC-targeting peptide (DCpep)	The NC8-pSIP409-dRVG could protect 60% of inoculated mice against deadly RABV challenge, even though the titers of RABV neutralizing antibody (VNA) were less than the threshold of 0.5 IU/mL	[152]
	cSN- KBLV	The full-length genome clone of SAD-B19 was constructed with the glycoprotein of Kotalahti Bat Lyssavirus	High levels of virus-neutralizing antibodies	[153]
Viral vector vaccines	rAAV-G	AAV-expressed G protein	Encouraged production of durable RVNAs in mice	[102]
	ChAd68 -Gp	Chimpanzee adenoviral vector-based rabies vaccine	Beagle dogs were completely protected even after receiving low doses of ChAd68-Gp intramuscularly and elicited potent immune responses	[137]
	ChAd155 -RG	Chimpanzee adenovirus vector serotype C expressing RABV-G	Augmented production of durable levels of RVNAs in mice, rabbits, and macaques	[154]
	VSV/RABV-GP	Replication-deficient vesicular stomatitis virus expressing RABV-G	Increased levels of RVNAs in mice	[110]
	VSV- RABV_G_	A replication- competent recombinant vesicular stomatitis virus expressing RABV-G	Asingle dose of VSV-RABVG intranasally resulted incomplete resistancetoRABV challenge in mice	[111]
	rNDV- R2B- FPCS- RVG	Mesogenic Newcastle disease virus (NDV) strain R2B expressing RABV-G	Generation of robust humoral and CMI responses in mice	[114]
	BHV-1 -ΔgE-G	Recombinant Bovine Herpes Virus Type 1-expressing RABV-G	Intramuscularly inoculated mice and cattle with no visible clinical signs had a protective level of RABV-specific virus-neutralizing antibody (VNA)	[115]
Intra Dermal Vaccines		A combined treatment protocol for rabies wound infiltration with eRIG and 5-day (0, 3, 7, 14, and 28) PEP regimen through an intradermal route in humans, dog, and cattle	Intradermal route was life saving in humans, cattle, and dogs	[155]
		Inactivated cell culture rabies vaccine administration viaSC, IM, and ID in dogs	ID was found to be safe and immunogenic in dogs	[156]
		Effectiveness of rabies pre-exposure prophylactic vaccines given withvarious methods in cattle	Appreciable levels of antibodies that neutralize the rabies virus (RVNA) through ID route in cattle	[157]

* Vaccine name was not designated.

**Table 2 vaccines-11-00756-t002:** Advantages and disadvantages of current rabies vaccines.

Vaccines	Advantages	Disadvantages
Modified live rabies vaccine	More immunogenic, long-lasting immunity with a single dose, economical	Potential reversion of virulence, more sensitive to changes in temperature, accidents of self-inoculation with MLV rabies vaccine pose a high risk to the vaccinator
Inactivated rabies vaccine	No potential reversion of virulence and safe	Low immunogenicity, requirement of repeated booster doses, expensive
Adjuvanted rabies vaccine	Immediate immune responses to produce higher levels and longer-lasting antibodies, elicit potent cellular and humoral immunity by enhancing antigen presentation to antigen-specific immune cells	Adverse side-effects of most of the adjuvant formulations
Protein/peptide vaccine	Safe vaccines, suitable in immunocompromised animals, peptides that block specific viral replication processes show promise as therapeutic vaccines	Poorly immunogenic and only capable of eliciting modest immune reactions, not cost-effective due to the requirement of extensive purification of the expressed protein
Nucleic acid vaccine	Helps in mass vaccinations, triggers both humoral and cellular immune responses	Poor immunogenicity, slow onset and modest induction of protective immune responses, multiple immunizations of high DNA doses are often required to produce enough antigen for optimal immune responses, potential risk of integration with the host genome
Genetically modifies vaccine	Safe, even in immunocompromised subjects, elicits high VNA titers and prompts early immune responses against RABV infection	Potential reversion of virulence, probable recombination with wild strains
Viral vector vaccines	Viral vectors carry PAMPs, which will elicit inflammatory responses needed for initiating adaptive immune responses, more immunogenic, useful as oral rabies vaccine candidates	Production of viral vectors is more complicated and costly, too reactogenic for use in humans, sometimes inadvertent infection of individuals with contact to the vaccine, immunity against the vector

## Data Availability

The data supporting reported results can be accessed in the references quoted respectively.

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
