# Peer review of "Developments in Rabies Vaccines: The Path Traversed from Pasteur to the Modern Era of Immunization"

_vaccines, 2023, doi:10.3390/vaccines11040756_

Round 1

Author Response

Specific comments:

The authors need to include below topics briefly to relevant places seperately:

  1. Immunity to natural rabies virus infection including humoral and cellular response

We thank the reviewer for this suggestion, accordingly the immune reaction in natural viral infection has been included in separate section in the manuscript (Page 19, 1st Paragraph,  Line 2).

  1. Immunity to rabies vaccines including humoral and cellular response

As suggested, included the information on immune reaction to rabies vaccines has been included in separate section in the manuscript (Page 19, 2nd Paragraph,  Line 1).

  1. Therapeutic approaches to rabies by using vaccines and antibodies.

Thank for this suggestion. The therapeutic approaches of vaccine as well the RIGs and siRNAhas been included under separate section in the manuscript (Page 19, 3rd  Paragraph,  Line 1).

Reviewer 2 Report

This article provides a good overview of the current situation with rabies vaccines which is useful for non-rabies experts like me as well as those more in touch with current rabies prevention studies..

Author Response

Sir

Thank you for approving the manuscript.

Reviewer 3 Report

This article presents a well-done paper on Pasteurs anniversary and a very relevant common thread in writing.

Author Response

(The authors gave the same response as above.)

Reviewer 4 Report

Overall Comments:

In this manuscript “Developments in Rabies vaccines: The path traversed from Pasteur to the modern era of immunization”, Krithiga and co-authors aimed to summary the path of rabies vaccine development, intended to highlight the drawbacks and to identify the gaps need to be filled in this field. To the reviewer’s knowledge, the current status of the article does not meet the quality requirements for publication in vaccines. This manuscript can be improved if authors follow the comment listed below.

Major issues:

1) First of all, the overall structure of the article is simple, its structure is not an outline of a systematic review. The author should include subheadings to better organize the content. In addition to introducing the epidemiology of rabies, the author should provide a detailed subsection on the virus's genome structure and particle morphology. A cartoon diagram should be included to help readers better understand the basic structure of the virus.

2) Introduction section: This section should be extensively revised. It should provide a systematic summary of relevant studies, point out the challenges and shortcomings of current detection platform, and highlight the novelty of this study. In addition, this section should be streamlined and condensed by combining some paragraphs and removing unnecessary redundancies. Some paragraphs should be the contents of an independent subtitle.

3) Also in introduction section, the authors should provide a comprehensive and systematic summary of the research progress in recent years.

4) It is suggested that the author should include a figure that demonstrates the transmission routes of rabies virus and the potential range of hosts.

5) The author should include a table summarizing the current research status of rabies vaccines and comparing the advantages and disadvantages of several types of vaccines.

6) In addition to summarizing the development status of rabies vaccines, the author should provide a section on future research directions to facilitate timely follow-up by researchers.

7) The overall writing quality is poor, extensive language revision is needed.

8) The abstract is poorly written, needs to reorganize the sentences and needs to be improved.

9) References: There is a lack of recent references in the manuscript, and the majority of the references are outdated. The author should update the reference section with recent publications.

Minor issues:

Line 18, Please pay attention to the format problem, such as line 18 and 61, delete the extra space. Check this issue throughout the manuscript.

Author Response

Overall Comments:

Major issues:

1) First of all, the overall structure of the article is simple, its structure is not an outline of a systematic review. The author should include subheadings to better organize the content. In addition to introducing the epidemiology of rabies, the author should provide a detailed subsection on the virus's genome structure and particle morphology. A cartoon diagram should be included to help readers better understand the basic structure of the virus.

Thank the reviewer for this suggestion. We have revised the manuscript to make its outline as that of a systematic review. As suggested, further subheadings have been included, details on rabies epidemiology and viral genome structure have been included in the introduction.

Epidemiology of rabies has been included in the introduction (Page no:2, 2nd Paragraph, Line no 4)

Viral genome structure and a diagramhave been included in the introduction (Page no: 3)

2) Introduction section: This section should be extensively revised. It should provide a systematic summary of relevant studies, point out the challenges and shortcomings of current detection platform, and highlight the novelty of this study. In addition, this section should be streamlined and condensed by combining some paragraphs and removing unnecessary redundancies. Some paragraphs should be the contents of an independent subtitle.

We thank the reviewer for this input. Accordingly, the introduction section is extensively revised. Challeges and short comings of the present detection platforms have been addressed (Page no:4, 3rd Paragraph, Line no 6). Furthermore, as desired by the reviewer, the novelty of this reviewhas been included in the introduction (Page no:5, 2nd Paragraph, Line no 1)

3) Also in introduction section, the authors should provide a comprehensive and Systematic summary of the research progress in recent years.

Provided the comprehensive and systematic summary of the recent research progress in a separate section(Page no:4, 4th Paragraph, Line no 1).

4) It is suggested that the author should include a figure that demonstrates the transmission routes of rabies virus and the potential range of hosts.

Included the figure demonstrating the transmission routes of rabies virus and potential range of targetshas been included in the introduction (Page no:3).

5) The author should include a table summarizing the current research status of rabies vaccines - and comparing the advantages and disadvantages of several types of vaccines.

Thank the reviewer for this suggestion. Included the table summarizing current research status of anti rabies vaccines (Page 22-26) and table comparing the advantages and disadvantages of various types of vaccines(Page 26-27).

6) In addition to summarizing the development status of rabies vaccines, the author should provide a section on future research directions to facilitate timely follow-up by researchers.

Provided a separate section on future research / perspectives for facilitating the researchers to follow up has been included (Page no:21, 1st  Paragraph, Line no 1)

7) The overall writing quality is poor, extensive language revision is needed.

The manuscript is revised to further improve the quality as well as language.

8) The abstract is poorly written, needs to reorganize the sentences and needs to be improved.

Reorganized the sentences in abstract and improved (Page 1, 1st Paragraph, Line no 1)

9) References: There is a lack of recent references in the manuscript, and the majority of the references are outdated. The author should update the reference section with recent publications.

We thank the reviewer for this suggestion. Recent references included.

Minor issues:

 Line 18, Please pay attention to the format problem, such as line 18 and 61, delete the extra space. Check this issue throughout the manuscript.

Thanks to reviewer for this observation. Taken care.

Round 2

Reviewer 4 Report

Overall Comments:

The author has adequately addressed all of the questions raised by the reviewers and made significant revisions, which warrant reconsideration for publication. However, the reviewer has the following concerns.

Major issues:

However, there are still some concerns that need to be addressed.

1)    Figure 1a and 1b should be merged into a single image, with the corresponding captions combined for presentation.

2)    Section 14: The author should merge the content into a coherent and logically organized paragraph to enhance the manuscript's overall structure.

3)    It was noted that all newly added images were copied from previously published works. This raises concerns about intellectual property rights and permission to reproduce these images. Do the authors get the permission?

Author Response

Major issues:

However, there are still some concerns that need to be addressed.

1)    Figure 1a and 1b should be merged into a single image, with the corresponding captions combined for presentation.

Thank the reviewer for this suggestion. Figures 1a and 1b have been merged into a single image as Figure1, with the combined corresponding captions (Page 2).

2)    Section 14: The author should merge the content into a coherent and logically organized paragraph to enhance the manuscript's overall structure.

We thank the reviewer for this input. The bullet points of the Future prospects of rabies vaccines in Section 14 were organized into a paragraph (Page no: 19, 2nd  Paragraph,  Line no 1).

3)    It was noted that all newly added images were copied from previously published works. This raises concerns about intellectual property rights and permission to reproduce these images. Do the authors get the permission?

We thank the reviewer for this suggestion. We have already received the permission from the respective authors for the incorporation of the new images in the manuscript as well the necessary citation for the source has been acknowledged.